

# The contents of essential and toxic metals in coffee beans and soil in Dale Woreda, Sidama Regional State, Southern Ethiopia

Yohannes Seifu Berego[1], Solomon Sorsa Sota[1], Mihret Ulsido[2] and Embialle Mengistie Beyene[3]

[1] Biology, Hawassa University, Hawassa, Sidama Region, Ethiopia
[2] Department of Water Supply and Environmental Engineering, Faculty of Biosystems and Water Resources Engineering, Institute of Technology, Hawassa University, Hawassa, Sidama Region, Ethiopia
[3] Environmental Health, Hawassa University, Hawassa, Sidama Region, Ethiopia

Corresponding author
Yohannes Seifu Berego,
yohannesseifu@hu.edu.et

## ABSTRACT

**Background**. For developing countries such as Ethiopia, coffee is a commodity of great economic, social, and environmental importance. No detailed investigations have been performed on the contents of essential and toxic metals in coffee beans and soil in this study area.

**Methods**. The levels of essential metals (Na, K, Ca, Zn, Mn, Cu, Co, Cr, Ni) and toxic elements (Pb and Cd) were investigated in coffee beans (coffee growing farmland and coffee washed plants) and soil samples (from farmland) using flame atomic absorption spectrometry (FAAS) and flame emission atomic spectroscopy. We selected six (20%) administrative units (kebele) with purposive sampling techniques based on their coffee production capacity in Dale Woreda for soil testing. After coffee sample preparation in a microwave system with $HNO_3$ and $H_2O_2$ reagents, the accuracy of the optimized procedure was evaluated by analysing the digest of the spiked samples. Soil samples were abridged with a slight revision of the EPA 3050B acid digesting method. ANOVA was used to determine the significant differences in the mean concentration of metal within coffee beans from farmland at the various sampled sites at the $p < 0.05$ significance level. To correlate the effect of one metal concentration on other metals in the coffee bean samples, Pearson correlation matrices were used.

**Results**. Calcium had the highest concentration ($1{,}355 \pm 18.02$ mg kg$^{-1}$) of macroelements in soil samples, followed by K ($681.43 \pm 1.52$ mg kg$^{-1}$). Similarly, Na ($111.63 \pm 0.35$ mg kg$^{-1}$), Cu ($49.96 \pm 0.99$ mg kg$^{-1}$), Co ($5.43 \pm 0.31$ mg kg$^{-1}$), Mn ($0.62 \pm 0.238$ mg kg$^{-1}$), Ni ($0.194 \pm 0.01$ mg kg$^{-1}$), and Zn ($0.163 \pm 0.007$ mg kg$^{-1}$) were detected among the microelements in the soil samples. Pb and Cr were not detected in all soil samples. Potassium (K) was found to have the highest concentration ($99.93 \pm 0.037$ mg kg$^{-1}$), followed by Ca ($17.23 \pm 0.36$ mg kg$^{-1}$), among the macroelements in coffee beans from farmers' farms. Similar to coffee beans from farmland, samples from washed plants also contained the highest K ($77.93 \pm 0.115$ mg kg$^{-1}$), followed by Ca ($4.33 \pm 0.035$ mg kg$^{-1}$). Metal levels in coffee bean samples from farmland are in the following order: K>Na>Ca >Mn>Cu> Ni>Zn. Metal levels were found to be K>Na>Ca >Mn>Cu> Zn>Ni in coffee beans from the washed plants. Co, Cr, Pb and Cd were no detected in all coffee bean samples. Except for calcium, potassium

and manganese, the levels of metals in coffee beans from farmland and washed plants were not significantly different at the 95% confidence level within a kebele.

**Conclusions**. We observed permitted levels of macro- and trace elements in coffee beans from farmlands and washed plants. Only in the soil samples are cadmium concentrations higher than those permitted for agricultural soil recommended by the WHO and FAO. Overall, there is no health danger linked with the use of coffee beans due to detrimental and trace heavy metals.

# INTRODUCTION

According to the International Coffee Organization *ICO (2021)*, coffee is the third most consumed beverage in the world after water and tea, and it is the world's second most traded commodity after oil. Various literature reports have noted the importance of moderate consumption of coffee on human health through the regulation of the level of sugar in the blood, as well as the prevention of diseases of the circulatory system and digestive system, cancers, and Parkinson's and Alzheimer's diseases (*Tan et al., 2007*; *Kotyczka et al., 2011*).

Coffee is a commodity of great economic, social and environmental importance in Ethiopia (*Gure, Chandravanshia & Godeto, 2017*). According to data from the Ethiopian Tea and Coffee Authority (2022), Ethiopia exported 300,000 metric tons of coffee in 2021, bringing $1.4 billion and accounting for almost 30% of all export revenue and supporting the livelihoods of over 25% of its population (*Tefera & Tefera, 2014*). According to *ICO (2021)*, Ethiopia ranks fifth in the world as a coffee producer and exporter after Brazil, Vietnam, Colombia and Indonesia and is the third largest consumer after Brazil and Indonesia (*ICO, 2021*). The Ethiopian output in 2020 reached 7.375 million bags (of 60 kg each) of processed coffee, nearly 4.3% of the global production (*ICO, 2021*).

Heavy metals by definition are metallic elements that have a high atomic weight and a much higher density at least five times that of water. They are stable elements, *i.e.,* They cannot be metabolized by the body and are bioaccumulative, *i.e.,* passed up the food chain to humans. Even at very low concentrations, they are highly toxic and can cause damaging effects (*Mohiuddin et al., 2011*). However, essential elements are those for which, even in small amounts, play important roles in healthy animals or plant life. Essential elements are vital for life (*Beasley et al., 2000*). Heavy metal contamination is a global hazard that began with the industrial revolution, resulting principally from farming practices through the application of organic fertilizers, minerals and pesticides to agricultural soil (*Carolin et al., 2017*; *Hejna, Gottardo & Baldi, 2018*), rapid industrialization and urbanization and the disposal of untreated and/or partially treated effluents from various industries. The pollution level has reached an alarming level in Ethiopia with increasing metal levels and deterioration of agricultural soil quality (*Ftsum & Abraha, 2018*). The stability of soil and water is impacted by pollution from heavy metals (*Kobielska et al., 2018*),

resulting in environmental (*Bello et al., 2018*) and public health problems (*Reyes, Torres & Gonzálesjimenez, 2016*).

Metal toxicity results when our body intakes excess amounts through supplements, food and water (*Ibrahim, Froberg & Wolf, 2006*). Heavy metals exhibit specific toxic effects on human beings and damage to the kidney, liver, lung, blood cells, and mental and central nervous systems (*Dorne, Kass & Bordajandi, 2011*; *Winiarska-Mieczan et al., 2021*). Chronic exposure can lead to a gradual increase in neurodegenerative processes, which is related to diseases such as multiple sclerosis and Alzheimer's disease (*FAO, 2011*; *Butt & Sultan, 2011*). Heavy metals are toxic even in very small amounts, such as arsenic, cadmium, chromium, nickel, and mercury which are classified as category one carcinogens, as they increase cancer risk in humans even with mild to moderate exposure (*Jaishankar, Tseten & Anbalagan, 2014*; *Kim, Kim & Seo, 2015*).

Recent research shows that some of our favorite coffee brews can be laced with contaminants, and many studies have been conducted worldwide to determine the levels of essential and toxic elements in coffee beans (*Gure, Chandravanshia & Godeto , 2017*; *Feleke et al., 2018*; *Rubio et al., 2019*; *Albals et al., 2021*; *Dubale, 2021*). Other research has been conducted to determine heavy metal accumulation in cacao beans (*Takrama et al., 2015*; *Bertoldi et al., 2016*; *Aguirre-Forero, Piraneque-Gambasica & Vasquez-Polo, 2020*). Previous studies have also assessed the levels of hazardous components in grains from various countries (*Meharg et al., 2013*; *Tatahmentan et al., 2020*). The transfer of metals from soil to coffee beans or other crop products, such as cacao grains, can be absorbed by plants, where they can either store them in the roots or move them into the shoots and grains (*Silva, Vitti & Trevizam, 2007*). When metals get into the coffee beans, they become sources of contamination for people, causing harmful health effects such as significantly reduced neurological and hepatic functioning, mutagenesis, and carcinogenesis (*Matés, Safe & Alonso, 2010*).

Sidama coffee beans are well known on the global market for their superior quality. Due to this, Sidama coffee bean prices and consumption have increased over the past few years (*Gelaw, 2019*). There have been previous studies on the determination of metals in coffee beans, but these studies had different scopes, such as coffee beans samples which were taken from farmer farms and coffee washing industries. To the best of our knowledge, no detailed investigations have been performed on the contents of essential and toxic metals in coffee beans and soil in Dale Woreda, Sidama Regional State. Moreover, the knowledge gap on the contents of essential and toxic metals in coffee beans and soil needs to be filled. Therefore, the aim of this study is to investigate the contents of essential and toxic metals in coffee beans and soil in Dale Woreda, Sidama Regional State, Southern Ethiopia.

## MATERIALS & METHODS

### Description of the study area

This study was carried out in Dale Woreda, Ethiopia. The geographic location of Woreda is between latitude 6°41′35″ north and longitude 38°21′17″ east. The capital town of Dale woreda is Yirgalem, which is located 45 km from Hawassa. The southern, western,

northern, and eastern borders of Dale are formed by Aleta Wendo and Chuko, Loka Abaya, Boricha, Shebedino, and Wensho. Dale Woreda's elevation along the Lake Abaya shoreline varies from 1,200 m above sea level to approximately 3,200 m at its westernmost point. Rivers include the Gidabo. Coffee is an important cash crop in Dale, with 17.38 square kilometers planted with this crop, which produced a total of 12.3 million kilograms of beans in 2020/21. Industry in this Woreda includes 51 coffee pulpers (*DWFEDO, 2021*). Farming chemicals are being utilized in agricultural operations more frequently, possibly to increase yields. The range of the average yearly temperature is between 9.6 °C and 29.2 °C. There are two rainy seasons in the area, with the first peak being in April–May and the second from August–October. The least amount of rain fell between November and February. The area receives 1,102 mm of rain on average yearly. Agroforestry practices appear to be the main features of the land use systems in the region. The main constituents of coffee are caffeine, tannin, fixed oil, carbohydrates, and proteins. It contains 2–3% caffeine, 3–5% tannins, 13% proteins, and 10–15% fixed oils.

## Sample collection

For this study, out of the 32 Kebele Administrations in Dale Woreda, only 20% were selected, which included six kebele. Coffee beans were collected from farmer's farms with supporting soil from six kebeles, Kege (SS1), Wenenata (SS2), Gane (SS3), Wondo (SS4), Bera (SS5), and Megara (SS6), of Dale Woreda using a grab sampling method. To obtain coffee bean samples, eight farmers from one kebele were chosen based on how much coffee they could produce. Next, a minimum of five coffee plants from a farm with a single farmer were used for sampling. Finally, the entire samples were homogenized to create a single sample of coffee beans that is indicative of a single kebele. The accumulated seize samples have been blended to form a composite sample to end up as the representative sample of the bulk. For sampling coffee beans, red cherries harvested during their peak season were carefully chosen and collected. To make pulping and grading easier, solely ripe crimson cherries have been collected, (*i.e.,* the entire cherry after harvest is first cleaned to separate the unripe, overripe and damaged cherries and to remove dirt, soil, twigs and leaves. Using stainless steel forceps, the soil samples were cleansed. The outer layers of the coffee cherry were also removed. Wash the green coffee beans that have been separated from their outer coat with tap water before washing them with deionized water to reduce contamination. The samples were sealed in polythene bags. Coffee washing plants, on the other hand, were chosen based totally on the following manner. In the six chosen kebeles, there were 13 coffee washing plants. One coffee washing plant/industry was purposively chosen from a single kebele. For this study, a total of six coffee processing plants were selected. All coffee processing industries in Dale Woreda used wet processing consisting of the following steps: sorting and cleaning, pulping, fermentation, coffee washing and drying. By submerging them in water, coffee cherries are sorted. The excellent, mature fruit will sink, while the bad, unripe fruit will float. Cherry skin and some pulp are removed by pressing the fruit mechanically through a screen in water. There will still be a substantial amount of pulp stuck to the bean that needs to be removed. This is accomplished by a more recent technique known as machine-assisted wet processing. The remaining pulp is

removed in the ferment-and-wash method of wet processing by causing the cellulose to breakdown while the beans are fermented with bacteria and then thoroughly washed. To obtain samples of coffee beans from washing plants, the washing company left the coffee beans to dry on beds after washing them, and we used those samples for sampling. All samples were taken in triplicate.

The soil samples were amassed from the root of the sample coffee plant from which the coffee beans were collected. Soil samples were collected from the floor 15 cm–25 cm under each sampled coffee plant (*Csuros & Csuros, 2002*). Because the study examined the uptake of vital and nonessential metals by coffee plants, samples were taken from the complete place where the plant's root system penetrates. Finally, the samples were placed into nonreacting polyethylene bags and carefully blended to make one composite pattern for each Kebele, which was then delivered to the laboratory.

## Preparation of coffee bean samples

On a firm, flat, easy surface, such as raised tables, the accumulated coffee cherries have been dried in direct sunlight. Drying was used to eliminate moisture from the coffee bean in a slow continuous technique, as it takes up to 4 weeks before the cherries were dried to the greatest moisture content, relying on the climate conditions. The beans are then removed from the dried husk. After grinding the dried coffee beans, 50 g was used for analysis. Finally, the powdered coffee bean samples were stored in polyethylene plastic luggage and stored in desiccators containing calcium chloride to maintain a constant dry weight until digestion" (*Mitra, 2003*). Samples taken from washing plants were also prepared in the same way as the beans from farmland.

## Preparation of soil samples

Any visible plant remnants were removed, and the soil samples were air dried and homogenized. The dried soil samples were ground and sieved using two mm nylon sieves. The total amount of soil samples gathered from a single Kebele furnished over 500 g of sieved soil, of which 50 g was used for chemical analysis. Before chemical analysis, the sieved soil pattern had been further dried in an oven at 50 °C for one and 1/2 h to make its moisture content material uniform. Finally, the samples were saved in sealed polythene and stored in desiccators containing calcium chloride to maintain a regular dry weight until digestion (*Mitra, 2003*).

## Optimization of digestion conditions

Different microwave digestion techniques were used in an effort to obtain a clear and colourless coffee digest solution that was adequate for FAAS analysis. $HNO_3$ and $H_2O_2$ volume, microwave digestion temperature, and digestion duration were the main factors. The three aforementioned factors were changed one at a time to create a total of 20 trials. The best digestion method was chosen based on the following factors: the clarity of the digests (solution free of residue and suspended materials), the smallest reagent volume, the shortest microwave digestion time, and the highest temperature. The optimum digestion conditions for coffee bean samples are presented in (Table 1).

**Table 1  Optimum microwave digestion conditions of coffee bean samples.**

| Step | 1 | 2 | 3 |
|---|---|---|---|
| Temperature °C | 150 | 200 | 45 |
| Time in minutes | 5 | 10 | 10 |
| Power-w | 80 | 85 | 0 |

## Digestion of coffee beans

For the evaluation of the concentration of elements using flame atomic absorption spectroscopy and flame emission atomic spectroscopy methods, samples of coffee must be solubilized. This sample preparation step mostly aims at reducing matrix effects originating from organic compounds and releasing elements in the form of their simple ions (*Castro et al., 2009*). A literature cited procedure used for the digestion of coffee powder (*Suseela, Bhalke & Kumar, 2001*) was used with a slight modification. A total of 0.3 g of bean powder was used and introduced to a 250 mL spherical bottom flask. The vessels were then put on the turntable of the microwave system after being predigested for 10 min in a fume hood with seven mL of $HNO_3$ 68% w/w conc. and two mL of $H_2O_2$ conc. Finally, the sample was digested under the optimum conditions. For each bulk sample, digestion was performed three times. In parallel to the digestion of the samples, the same method was used to digest a reagent blank while maintaining all other digestion settings. Six blanks were digested for bean samples. The digest used was allowed to cool to room temperature LaCl3.7H2O (0.1%, w/v) was introduced to the digested solution to eliminate the chemical interference of Ca and Na ions. and the solution was then brought to the mark (25 mL) with deionized water. Triplicate digestions were carried out for each sample. The solutions were stored in the refrigerator until analysis.

## Digestion of soil samples

For digestion of soil samples, the EPA 3050B (*U.S. EPA, 1996*) method was applied. The system used for the digestion of the soil sample was as follows: first, a 1:1 ratio of $HNO_3$ and $H_2O$ (deionized) was prepared. Ten milliliters was added to a digestion vessel containing 500 mg of the dried and sieved soil sample. Finally, after completion of digestion with $HNO_3$, $H_2O_2$ and HCl, the mixture was allowed to cool and filtered *via* Whatman No. 42 filter paper, and the ensuing clear light yellow answer was made up to 50 mL with deionized water (*U.S. EPA, 1996*). Reagent blanks were additionally prepared and digested with the same procedure as that of the soil sample. All the solutions were stored in tightly capped polyethylene bottles and stored in a refrigerator until evaluation (*U.S. EPA, 1996*).

## Calibration procedure and determination of metals

For the purpose of determining the metal concentrations in soil and coffee bean sample solutions, calibration curves were prepared. The calibration curves for each metal were created using diluted stock standard solutions containing 1,000 mg/L each of the following metals: Ca, Cd, Co, Cu, Cr, K, Mn, Na, Ni, Pb and Zn in 2% $HNO_3$. The contents of the metals in coffee bean and soil samples were determined by using FAAS and flame emission atomic spectroscopy. To avoid loss through ionization, the concentrations of Na and K

were determined by the emission mode of the instrument. The same analytical approach was used to determine the elements in blank solutions, and three replicate determinations for each metal were conducted.

## Method validation

Method validation is the process used to confirm by examination and provision of objective evidence that the particular requirements for a specific intended use are fulfilled. Results from method validation can be used to judge the quality, reliability and consistency of analytical results; it is an integral part of any good analytical practice (*Ajay & Rohit, 2012*; *Magnusson & Örnemark, 2014*). Since there were no approved standard reference materials available, the approach was validated using spiking. Four flasks containing samples of coffee beans were spiked. In four different flasks, 0.3 g of coffee bean powder sample was taken. Two hundred milliliters of 1,000 mg/L Ca was spiked into the first flask. The second flask was spiked with 1,500 L of the same K concentration as the first flask. Twenty-five liters of 10 mg/L Cu, Mn, and Zn were spiked into the third flask. In the fourth flask, 15 µL of 10 mg/L Ni, Co and Cr was spiked. All spiked samples were digested in triplicate following the optimal digestion procedure developed for coffee bean samples.

## Analysis of samples for metals

After the parameters such as the burner and lamp alignment, slit width, and wavelength adjustment were optimized for the maximum signal intensity of the instrument, the elements were determined as previously described in *Amente (2016)*, specifically by using flame atomic absorption spectroscopy and flame emission atomic spectroscopy with an external calibration curve. Each metal's hollow cathode lamp was put into the atomic absorption spectrophotometer, which allowed the solution to be inhaled into the flame one at a time. To avoid loss *via* ionization, the attention of Na and K was decided with the aid of flame emission atomic spectroscopy. Calibration curves were organized to determine the concentration of the metals in coffee bean and soil pattern solutions. Calibration standards were made for each of the following metals using stock standard solutions (Buck Scientific Instruments, LCC, Norwalk, CT, USA) containing 1,000 mg/L of each metal. The elements Na, K, Ca, Zn, Mn, Cu, Co, Cr, Ni, Pb, and Cd were among those used. A 10 mg/L intermediate standard was produced from them.

## Calculation of bioaccumulation

The bioconcentration factor (BCF) is the ratio of the heavy metal concentration in the edible part of the plant to the heavy metal concentration in the soil sample (*Rattan et al., 2005*; *Sharma, Nagpal & Kaur, 2018*). Accordingly, heavy metal transfer from soil to plant was calculated by the formula used by (*Kachenko & Singh, 2004*) and given in Eq. (1).

$$\text{BCF} = \frac{\text{heavy metal content in plant}}{\text{heavy metal content in respective soil}} \tag{1}$$

A BCF value greater than 1 indicates that the plant is a potential accumulator for the metal being considered for analysis (*Barman et al., 2000*).

## Data analysis

An analysis was performed using SPSS version 24, and ANOVA was used to determine the significant differences in the mean concentration of metal within coffee beans from farmland at the various sampled sites at the $p < 0.05$ significance level. Additionally, t tests were performed to check whether there was a difference in metal concentrations between coffee beans from farmland and washing industries. Finally, to correlate the effect of one metal concentration on the concentration of the other metal in the coffee bean samples, Pearson correlation matrices using the correlation coefficient (r) for the samples were used.

## RESULTS

### Metal concentration in soil samples

The highest mean Ni concentration ($0.194 \pm 0.009$ mg kg$^{-1}$) was recorded in soil sampled from Kege Kebele (**SS1**), and the lowest ($0.172 \pm 0.002$ mg kg$^{-1}$) was recorded in soil sampled from Bera Kebele (**SS5**). (Table 2). The highest mean Zn concentration ($0.163 \pm 0.007$ mg kg$^{-1}$) was recorded in soil sampled from SS1, and the lowest ($0.141 \pm 0.001$ mg kg$^{-1}$) was recorded in soil from SS3. The mean Co concentration was reported at two sites: $5.43 \pm 0.305$ mg kg$^{-1}$ in soil samples from SS1 and $4.27 \pm 0.20$ mg kg$^{-1}$ at SS2. The highest mean Cu concentration ($49.96 \pm 0.99$ mg kg$^{-1}$) was recorded in soil sampled from SS3, and the lowest ($22.34 \pm 0.25$ mg kg$^{-1}$) was recorded in soil from SS4.

The cadmium levels in the soil are shown in Table 2. The mean concentration of Cd varied between $2.38 \pm 0.044$ and $3.36 \pm 0.1$, with the highest concentration in the sample from SS1 and the lowest concentration from SS4. Calcium had the highest concentration ($1,355 \pm 18.02$ mg kg$^{-1}$) of macro elements in all soil samples, followed by K ($681.43 \pm 1.52$ mg kg$^{-1}$) and Na ($111.63 \pm 0.35$ mg kg$^{-1}$). Similarly, Cu ($49.96 \pm 0.99$ mg kg$^{-1}$) was detected in greater abundance among the microelements, followed by Mn ($0.62 \pm 0.238$ mg kg$^{-1}$) and Zn ($0.163 \pm 0.007$ mg kg$^{-1}$). Co>Cd>Ni was assigned to the remaining trace metals (Table 2). In soil samples from six kebeles, metals such as Pb and Cr were not detected. Kege Kebele soil sample analysis showed that Ca ($1355 \pm 18.02$ mg kg$^{-1}$) was the highest of the examined metals. K and Na were traced in the highest concentration next to Ca with concentrations of $673 \pm 2.65$ mg kg$^{-1}$ and $111.63 \pm 0.35$ mg kg$^{-1}$, respectively.

Of the analysed microelements, Cu ($42.66 \pm 2.52$ mg kg$^{-1}$) was found to be in the highest amount, followed by Co ($5.33 \pm 0.305$ mg kg$^{-1}$) and Mn ($0.62 \pm 0.238$ mg kg$^{-1}$). Similarly, the levels of Cd, Ni and Zn were $3.1 \pm 0.1$ mg kg$^{-1}$, $0.191 \pm 0.009$ mg kg$^{-1}$ and $0.161 \pm 0.007$ mg kg$^{-1}$, respectively. The average metal concentrations in the kege soil were determined to be in the following order: Ca>K>Na>Cu>Co>Cd>Mn>Ni>Zn (Table 2).

Pearson's correlation coefficient was used to examine the correlations between the contents of different elements in soil. The analysis revealed that there were significant ($p < 0.01$, $p < 0.05$) positive correlations between heavy elements: Cd-Zn ($r = 0.511$), Cd-Ca ($r = 0.507$), Cd-Na ($r = 0.81$), Cd-K ($r = 0.57$), Ni-Co ($r = 0.882$), Zn-Co ($r = 0.469$), Cu-Ca ($r = 0.537$), Ca-Na ($r = 0.652$), and Na-K ($r = 0.69$); all had positive correlations (Table 3). Other correlations were insignificant, such as Cd with Ni, Co, Cu, Mn, and K.

Berego et al. (2023), *PeerJ*, DOI 10.7717/peerj.14789

**Table 2 Mean metal concentrations in soil samples (mean standard deviation).**

|  | Ca | Cd | Co | Cu | K | Mn | Na | Ni | Zn |
|---|---|---|---|---|---|---|---|---|---|
| SS1 | 1,355 ± 18.02 | 3.36 ±0.10 | 5.43 ±0.31 | 42.46 ± 2.52 | 673 ± 2.65 | 0.62 ±0.24 | 111.63 ± 0.35 | 0.194 ± 0.01 | 0.163 ± 0.01 |
| SS2 | 1,538.67 ± 3.20 | 2.53 ±0.06 | 4.27 ± 0.20 | 47.48 ± 0.58 | 403.47 ± 1.53 | 0.5 ± 0.001 | 26.52 ± 0.04 | 0.181 ± 0.001 | 0.146 ± 0.002 |
| SS3 | 445 ± 0.99 | 2.67 ±0.06 | ND | 49.96 ± 0.99 | 603 ± 0.10 | 0.467 ±0.02 | 26.77 ± 0.02 | 0.191 ± 0.001 | 0.141 ± 0.001 |
| SS4 | 346.67 ± 2.51 | 2.38 ± 0.04 | ND | 22.34 ± 0.25 | 604.61 ± 0.58 | 0.48 ± 0.002 | 13.37 ± 0.032 | 0.181 ± 0.001 | 0.155 ± 0.005 |
| SS5 | 1,352.67 ± 2.50 | 2.93 ± 0.02 | ND | 47.7 ± 1.15 | 681.43 ± 1.52 | 0.49 ± 0.001 | 112 ± 0.10 | 0.172 ± 0.002 | 0.153 ± 0.002 |
| SS6 | 1,401 ± 1.10 | 3.01 ± 0.006 | ND | 45.26 ± 1.52 | 680.43 ± 0.57 | 0.50 ± 0.001 | 110.3 ± 0.10 | 0.178 ± 0.001 | 0.146 ± 0.005 |
| MPL | – | 3.0 | 50.0 | 300.0 | – | – | – | 50.0 | 1,000.0 |

**Notes.**

ND, not detected; MPL, Maximum Permissible Limit for Agricultural soils according to *FAO & WHO (2019)*.
**Table 3  Correlation analysis of metals of soil in the selected Kebele of Dale Woreda.**

|     | Cd | Ni | Zn | Cu | Ca | Mn | Na | K |
|-----|-----|-----|-----|-----|-----|-----|-----|-----|
| Cd | 1 | | | | | | | |
| Ni | 0.337 | 1 | | | | | | |
| Zn | 0.511* | 0.241 | 1 | | | | | |
| Co | 0.359 | 0.482* | 0.469* | | | | | |
| Cu | 0.363 | 0.035 | −0.421 | 1 | | | | |
| Ca | 0.507* | −0.285 | 0.144 | 0.537* | 1 | | | |
| Mn | 0.135 | 0.155 | 0.334 | −0.014 | 0.266 | 1 | | |
| Na | 0.81** | −0.161 | 0.396 | 0.369 | 0.65** | 0.325 | 1 | |
| K | 0.57* | 0.012 | 0.379 | −0.075 | −0.096 | 0.151 | 0.69** | 1 |

**Notes.**
  *Correlation is significant at the 0.05 level (2-tailed).
  **Correlation is significant at the 0.01 level.

Ni was found to be nonsignificant with Zn, Cu, Ca, Mn, and K. Zn was also found to be non-significant with Cu, Ca, Mn, Na and K. Cu was found to be non-significant with Mn. Na. Similarly, Ca was found to be non-significant with Mn and K. Likely Mn was found to be non-significant with Na and K, indicating that the concentration of one element may not have an impact on the concentrations of other elements in the study area. The positive correlation between Cd and Zn, Ca, Na and K indicates a correlation among the metals in coffee beans. The rise in levels of Cd increases the tendency of Zn, Ca, Na and K to increase.

## Metals in coffee bean samples from farm land

Potassium (K) was found to have the greatest concentration ($99.93 \pm 0.04$ mg kg$^{-1}$) among the macro-elements detected in coffee beans from all sampling farm land sites (Kege, Wenenata, Gane, Wondo, Bera, and Magara). The findings of the present study revealed that Na and Ca were also detected in significant amounts in coffee bean samples from farmers' farms, with concentrations of $22.04 \pm 0.042$ mg kg$^{-1}$ and $17.23 \pm 0.36$ mg kg$^{-1}$, respectively. The highest concentration observed for manganese in this study was $0.927 \pm 0.004$ mg kg$^{-1}$ (Table 4).

The highest concentration of Ni ($0.074 \pm 0.003$ mg kg$^{-1}$) was reported in coffee beans from farmland. The zinc concentration in the samples analysed ranged between $0.054-0.076$ mg kg$^{-1}$. The copper concentration in the present study ranged between $0.14-0.28$ mg kg$^{-1}$. In coffee beans of the selected study site, Pb, Co, Cr, and Cd were not found (Table 5).

In coffee beans from Kege farmers' farms, Mn was determined to be the most abundant minor element, followed by Cu, Zn and Ni. Likewise, the levels of Cu ($0.22 \pm 0.0026$ mg kg$^{-1}$) and Zn ($0.073 \pm 0.003$ mg kg$^{-1}$) were higher than that of Ni ($0.055 \pm 0.004$ mg kg$^{-1}$). Except for Pb, Cd, Co and Cr, which were not detected in coffee beans, all kinds had equal minor element concentrations. In farmer's coffee beans from Wenenta, Wondo, Gane, Bera, and Magara Kebele, nearly the matching dissemination pattern of elements was detected as in Kege Kebele (Table 5).

**Table 4** Concentration of essentials metal concentration (mean ± SD, *n* = 3, mg kg-1) in coffee bean sample from farm land in Dale Woreda, Ethiopia, 2021.

|  |  | K | Ca | Na | Mn |
|---|---|---|---|---|---|
| Kege |  | 99.93 ± 0.037 | 15.15 ± 0.614 | 22.04 ± 0.042 | 0.927 ± 0.004 |
| Wenenata |  | 69.06 ± 0.047 | 17.23 ± 0.36 | 5.73 ± 0.049 | 0.105 ± 0.003 |
| Gane |  | 80.587 ± 0.08 | 3.05 ± 0.04 | 6.927 ± 0.038 | 0.082 ± 0.003 |
| Wondo |  | 78.10 ± 1.081 | 3.96 ± 0.063 | 3.10 ± 0.10 | 0.088 ± 0.003 |
| Bera |  | 69.36 ± 0.55 | 16.11 ± 0.11 | 5.73 ± 0.049 | 0.105 ± 0.003 |
| Magara |  | 99.27 ± 0.714 | 15.33 ± 0.189 | 21.077 ± 0.99 | 0.093 ± 0.004 |
| MPL | Medicinal plant | 32,500 | – | – | – |
|  | Edible plant | – | – | – | 2 |

Notes.
MPL, Maximum Permissible Limit for medicinal and edible plants according to *FAO/WHO (1993)*.

**Table 5** Concentration of toxic metal concentration in coffee bean sample from farmer's farms in Dale Woreda, Ethiopia, 2021.

|  |  | Ni | Zn | Co | Cu | Cr | Pb | Cd |
|---|---|---|---|---|---|---|---|---|
| Kege |  | 0.055 ± 0.004 | 0.073 ± 0.003 | ND | 0.22 ± 0.0026 | ND | ND | ND |
| Wenenata |  | 0.053 ± 0.003 | 0.065 ± 0.004 | ND | 0.24 ± 0.002 | ND | ND | ND |
| Gane |  | 0.074 ± 0.003 | 0.055 ± 0.002 | ND | 0.28 ± 0.0037 | ND | ND | ND |
| Wondo |  | 0.052 ± 0.005 | 0.07 ± 0.0017 | ND | 0.14 ± 0.0020 | ND | ND | ND |
| Bera |  | 0.057 ± 0.003 | 0.067 ± 0.002 | ND | 0.234 ± 0.01 | ND | ND | ND |
| Magara |  | 0.057 ± 0.008 | 0.07 ± 0.001 | ND | 0.22 ± 0.0005 | ND | ND | ND |
| MPL | Medicinal plant | 1.63 | No MPL | No MPL | 20 | 2 | 10 | 0.3 |
|  | Edible plant | No MPL | 27.4 | No MPL | 3 | 0.02 | 0.43 | 0.21 |

Notes.
ND, Not detected; MPL, Maximum Permissible Limit for medicinal and edible plants according to *FAO/WHO (1993)*.

## Bio-concentration factor of trace metals from soil to coffee bean/Transfer factor

Mn had the highest bio-concentration factor/transfer factor (1.495) among the elements examined. However, among the examined samples, Cu had the lowest transfer factor (0.0048), which is at the Magara locations (SS6). In Kege Kebele (SS1), the overall transfer pattern for trace metals was Mn >Zn >Ni >Na>K >Ca >Cu (Table 6). This demonstrates unequivocally that Mn bioaccumulation factors were higher in coffee samples than for other metals.

## Metals in coffee bean samples from the washing industry

The beans from Megara coffee washing industries, similar coffee beans from farmlands, had the maximum amount of K ($77.93 \pm 0.115$ mg kg$^{-1}$), followed by Na ($10.47 \pm 0.058$ mg kg$^{-1}$) and Ca ($3.55 \pm 0.114$ mg kg$^{-1}$). In the washing industry's beans, Mn ($0.92 \pm 0.001$ mg kg$^{-1}$) was the highest accumulated trace metal, followed by Cu ($0.277 \pm 0.011$ mg kg$^{-1}$) and Zn ($0.094 \pm 0.004$ mg kg$^{-1}$). Other critical trace metal concentrations found in coffee beans were Ni ($0.074 \pm 0.003$ mg kg$^{-1}$). The results show that the concentrations of Co, Cr, Pb and Cd were not detected. The dissemination of elements in coffee beans from all six Kebele washing plant trails showed the same trend, as illustrated in (Table 7). One-way

**Table 6  Bio-concentration factor (BFC) of heavy metals analyzed in coffee bean grown by Dale Woreda and its farm land.** A BCF number greater than 1 implies that the plant is a potential metal accumulator, as shown by the bold value.

| Sampling area | Transfer of trace metals | | | | | | |
|---|---|---|---|---|---|---|---|
| | K | Ca | Na | Mn | Ni | Zn | Cu |
| Kege | 0.149 | 0.011 | 0.197 | **1.495** | 0.284 | 0.448 | 0.0052 |
| Wenenata | 0.171 | 0.011 | 0.216 | 0.21 | 0.293 | 0.445 | 0.0051 |
| gane | 0.134 | 0.0068 | 0.258 | 0.176 | 0.387 | 0.390 | 0.0056 |
| wondo | 0.129 | 0.0114 | 0.232 | 0.183 | 0.287 | 0.452 | 0.0063 |
| Bera | 0.102 | 0.0119 | 0.051 | 0.214 | 0.331 | 0.438 | 0.0049 |
| Megara | 0.146 | 0.0109 | 0.191 | 0.186 | 0.320 | 0.479 | 0.0048 |

ANOVA showed that significant differences were found in the metal concentrations ($p > 0.05$) at 95% confidence levels for Ca, Cu, Mn and Ni in coffee samples collected from all washing plants. The maximum allowable limits of heavy metals in coffee beans were not found in the literature. As a result, standards for other foods and herbal plants were used for comparison (Table 8). The results in Table 9 show that the element content found is more or less analogous to levels published in the literature (Table 9).

Except for Ca and K in Kege Kebele, Mn in Bera Kebele, Ca and K in Wenenata, K in Gane, Wodo and Megara Kebele, all other measured elements were not significantly different ($p > 0.05$) in coffee beans from farmlands and washing plants, according to the findings (Table 10).

## DISCUSSION

The concentration of Ni ($0.194 \pm 0.009$ mg kg$^{-1}$) in the soil sampled in the present study was much lower ($8.33 \pm 0.55$ mg kg$^{-1}$) than that reported in Ethiopian farms where coffee is grown (*Dubale, 2021*). This difference might be due to the use of diverse farming practices, as well as geographical, meteorological and geological differences between the study areas. Likewise, it may be more common in the Gedeo zone to apply livestock manure to agricultural soil, which could be the cause of the soil's high Ni concentration (*Basta, Ryan & Chaney, 2005*; *Hejna, Gottardo & Baldi, 2018*). Similarly, the findings of the present study were much lower than reported Ni ($5.1 \pm 3.8$ mg kg$^{-1}$) in agricultural land in Kenya (*Ndungu et al., 2019*), reported Ni (133.1 mg kg$^{-1}$) in Brazil farms where coffee is grown (*Tezottoa et al., 2012*) and reported Ni ($1.92 \pm 1$ mg kg$^{-1}$) from Brazilian coffee cultivated areas (*Santos et al., 2009*). However, the finding of the present study was much higher (0.05 mg kg$^{-1}$) than that reported in the Western Ghats, India (*Raghavendra & Venkatesha, 2020*). This is due to the use of diverse farming practices, as well as geographical, meteorological and geological differences between the study areas. Moreover, the application of livestock manure to agricultural soil is a common practice in the study areas and could be a possible reason for the high Ni concentration in the soil (*Basta, Ryan & Chaney, 2005*). The Ni levels analyzed were below the permissible limit set by *FAO & WHO, 2019* (50.00 mg kg$^{-1}$); therefore, these soils are free of contamination.

Berego et al. (2023), *PeerJ*, DOI 10.7717/peerj.14789

**Table 7** Mean concentration (mean ± SD, $n = 3$, ppm) of elements in coffee bean sample from washing industry in Dale Woreda, Ethiopia, 2021.

| | Ca | Cd | Co | Cu | K | Mn | Na | Ni | Zn |
|---|---|---|---|---|---|---|---|---|---|
| SS1 | 3.733 ± 0.03[a] | ND | ND | 0.243 ± 0.003[a] | 75.81 ± 0.026[a] | 0.92 ± 0.001[a] | 10.30 ± 0.10[b] | 0.074 ± 0.003[a] | 0.054 ± 0.004[d] |
| SS2 | 2.816 ± 0.021[a] | ND | ND | 0.191 ± 0.001[a] | 61.39 ± 0.026[c] | 0.08 ± 0.003[a] | 15.04 ± 0.04[a] | 0.074 ± 0.003[a] | 0.063 ± 0.004[c] |
| SS3 | 3.71 ± 0.015[a] | ND | ND | 0.254 ± 0.004[a] | 71.07 ± 0.049[a] | 0.09 ± 0.004[a] | 3.43 ± 0.043[c] | 0.054 ± 0.004[a] | 0.075 ± 0.002[b] |
| SS4 | 4.33 ± 0.035[a] | ND | ND | 0.146 ± 0.004[a] | 68.30 ± 0.093[b] | 0.09 ± 0.003[a] | 1.43 ± 0.035d | 0.073 ± 0.002[a] | 0.094 ± 0.004[a] |
| SS5 | 2.81 ± 0.017[a] | ND | ND | 0.191 ± 0.001[a] | 61.59 ± 0.356[c] | 0.08 ± 0.005[a] | 15.04 ± 0.041[a] | 0.052 ± 0.001[a] | 0.067 ± 0.004[c] |
| SS6 | 3.55 ± 0.114[a] | ND | ND | 0.277 ± 0.011[a] | 77.93 ± 0.115[a] | 0.09 ± 0.001[a] | 10.47 ± 0.058[b] | 0.071 ± 0.001[a] | 0.053 ± 0.002[d] |
**Table 8 Comparison of current results for coffee beans from farms and the washing industry with FAO/WHO, different organizations, and nations' maximum permissible values for metals.**

| Elements | Present study | | MPL (ppm) | Type of plant | Refrences |
|---|---|---|---|---|---|
| | CBFF | CBWI | | | |
| Ni | 0.05–0.08 | 0.05–0.08 | 1.63 | Edible plant | *FAO/WHO (1993)* |
| | | | No MPL | Medicinal plant | *WHO (2005)* |
| | | | 50 | Grain | USAD (2000) |
| Zn | 0.054–0.076 | 0.051–0.09 | No MPL | Medicinal plant | *WHO (2005)* |
| | | | 100 | Beans | USAD (2000) |
| | | | 27.4 | Edible plant | *FAO/WHO (1993)* |
| Co | ND | ND | No MPL | Medicinal plant | *WHO (2005)* |
| | | | 40 | In food | *FAO/WHO (1993)* |
| Cu | 0.14–0.29 | 0.14–0.28 | 3 | Edible plant | *FAO/WHO (1993)* |
| | | | 20 | Medicinal plant | *WHO (2005)* set by Singapore |
| | | | 150 | Medicinal plant | *WHO (2005)* set by China |
| Cr | ND | ND | 2 | Medicinal plant | *WHO (2005)* |
| | | | 0.02 | Edible plant | *FAO/WHO (1993)* |
| Ca | 3.0–17.27 | 2.80–4.37 | – | – | |
| Mn | 0.08-0.11 | 0.07–0.10 | No MPL | Medicinal plant | *WHO (2005)* |
| | | | 2 | Edible plant | *FAO/WHO (1993)* |
| Na | 3.0–22.1 | 1.40–15.09 | – | – | |
| K | 69.0–99.97 | 61.4–78 | 32,500 | Medicinal plant | *FAO/WHO (1993)* |
| Pb | ND | ND | 0.43 | Edible plant | *FAO/WHO (1993)* |
| | | | 10 | Medicinal plant | *WHO (2005)* |
| Cd | ND | ND | 0.21 | Edible plant | *FAO/WHO (1993)* |
| | | | 0.3 | Medicinal plant | *WHO (2005)* |

**Notes.**
CBFF, coffee beans from farms; CBWI, Coffee beans washing industry; ND, Not detected; MPL, Maximum Permissible Limit.

The findings of the present study revealed that the concentration of zinc in the sampled soil was lower ($52.5 \pm 6$ mg kg$^{-1}$) than that reported in Brazil (*Santos et al., 2009*), and the reported Zn ($83.0 \pm 33.5$ mg kg$^{-1}$) concentration (*Tezottoa et al., 2012*), reported Zn ($10.0 \pm 3.1$ mg kg$^{-1}$) concentrations in agricultural land in Kenya (*Ndungu et al., 2019*) and Zn ($47.14 \pm 2.51$ mg kg$^{-1}$) from the Gedeo Zone, Ethiopia (*Dubale, 2021*). On the other hand, the present study found Zn levels much higher than those reported ($0.05$ mg kg$^{-1}$) in the Western Ghats Region, India (*Raghavendra & Venkatesha, 2020*). These changes in the concentration of heavy metals could be explained by the regular practice in the study area of applying livestock manure to agricultural soil (*Basta, Ryan & Chaney, 2005*; *Hejna, Gottardo & Baldi, 2018*). The mean concentration of Zn in soils from all sample sites was below the permissible limit set by *FAO & WHO (2019)* ($1,000$ mg kg$^{-1}$). This indicates that the soil from all sample sites is safe for agriculture with regard to zinc pollution.

The highest mean Co concentration was in the present study ($5.43 \pm 0.305$ mg kg$^{-1}$). Contrary to other Ethiopian research findings, the amount of cobalt in the present study was lower than the national average ($11.66 \pm 0.78$ mg kg$^{-1}$) concentration obtained in the Gedeo zone, Ethiopia (*Dubale, 2021*). However, the findings of the present study were

Berego et al. (2023), *PeerJ*, DOI 10.7717/peerj.14789

**Table 9** Metal concentrations in coffee beans compared to some of the literature values.

| | (*Onianwa et al., 1999*) Coffee beverage | (*Suseela, Bhalke & Kumar, 2001*) coffee powders | (*Silva et al., 2017*) roasted and ground coffee | (*Gure, Chandravanshia & Godeto , 2017*) in green coffee | (*Adler, Nkedzarek & Tórz, 2019*) in green coffee | (*Omer, Labib & Zafar, 2019*) in green coffee | (*Dubale, 2021*) in Ethiopia | | Present study | |
|---|---|---|---|---|---|---|---|---|---|---|
| | | | | | | | CBFF | CBWI | CBFF | CBWI |
| Ni | 0.04–2.58 | NR | 0.03–1.95 | <0.04–2.5 | 0.415 ± 0.04 | 0.0–0.25 | 1.66–2.43 | 1.56–2.32 | 0.05–0.08 | 0.05–0.08 |
| Zn | 4–14 | 2–9 | 5.53–55.8 | 4–21 | 3.6 ± 0.67 | 0.0–4.59 | 8.74–12.7 | 9.41–13.0 | 0.054–0.076 | 0.051–0.09 |
| Co | 0.1–14 | NR | NR | 2.6–8.4 | NR | NR | 2.47–2.86 | 2.31–2.75 | ND | ND |
| Cu | 2–9 | 0.4–16 | 0.7–17.18 | 11–23 | 14 ± 6.43 | NR | 23.4–28.5 | 23.1–28.2 | 0.14–0.29 | 0.14–0.28 |
| Cr | 0.89–6.98 | 0.4–1.00 | 0.03–0.10 | 0.21–0.28 | NR | NR | 1.04–1.92 | 0.94–1.90 | ND | ND |
| Ca | NR | 869–1171 | NR | 710–1250 | 789 ± 132.23 | 6.76–32.1 | 1037–1253 | 1090–1270 | 3.0–17.27 | 2.80–4.37 |
| Mn | NR | 7–13 | 9.81–39.8 | 13–19 | NR | 0.48–28.7 | 17.3–23.6 | 17.2–22.6 | 0.08–0.11 | 0.07–0.10 |
| Na | NR | NR | NR | NR | 18.6 ± 11.31 | 6.8-564.7 | NR | NR | 3.0–22.1 | 1.40-15.09 |
| K | NR | 14000–29000 | NR | 13010–17000 | 19898 ± 445.48 | 21.31–427.84 | 14631–15043 | 14602–14980 | 69.0–99.97 | 61.4–78 |
| Pb | 0.09–0.91 | NR | 0.03–1.58 | <0.05 | 0.076 ± 0.0956 | 0.0–23.88 | ND | ND | ND | ND |
| Cd | 0.02–0.31 | NR | 0.03–0.10 | ND | 0.015 ± 0.0005 | 0.00–8.01 | ND | ND | ND | ND |
**Table 10   T –test among coffee beans from farmer's farms and washing industries.**

|     | t | Sig. | Mean Diff. | Std. Error Diff | 95% CI | |
| --- | --- | --- | --- | --- | --- | --- |
|     |   |   |   |   | Lower | Upper |
| Ni | −1.91 | 0.064 | −0.00594 | 0.003109 | −0.01226 | 0.000373 |
| Zn | −0.39 | 0.694 | −0.00149 | 0.003757 | −0.00912 | 0.006146 |
| Cu | 0.58 | 0.567 | 0.008444 | 0.014612 | −0.02125 | 0.03814 |
| Ca | 5.77 | 0.01 | 8.310556 | 1.44024 | 5.383635 | 11.23748 |
| Mn | 2.28 | 0.029 | 0.006111 | 0.002682 | 0.00066 | 0.011562 |
| Na | 0.66 | 0.517 | 1.482222 | 2.264297 | −3.11938 | 6.083828 |
| K | 3.89 | 0.01 | 13.36944 | 3.43744 | 6.383726 | 20.35516 |

**Notes.**

CI, Confidence interval.

higher than the Co ($3.57 \, \text{mg kg}^{-1}$) concentration reported in Kaduna State, Nigeria (*Oladeji & Saeed, 2015*). This could be due to anthropogenic sources of contamination, such as the widespread use of manure, herbicides, fungicides, and fertilizers, which are high in Co, as suggested by *Yasmeen et al. (2010)*.

The concentration of Cu ($49.96 \pm 0.99 \, \text{mg kg}^{-1}$) in the soil sampled in the present study was much lower ($76.9 \pm 1.34 \, \text{mg kg}^{-1}$) than that reported in Ethiopian farms where coffee is grown (*Dubale, 2021*). However, the concentration of Cu reported by the present study is much higher ($7.10 \pm 0.30 \, \text{mg kg}^{-1}$) than that reported in Brazilian coffees (*Santos et al., 2009*), reported Cu ($36.8 \pm 5.9 \, \text{mg kg}^{-1}$) concentration in Brazil farms where coffee is grown (*Tezottoa et al., 2012*), and Cu ($0.26 \pm 0.17$) in agricultural land in Kenya (*Ndungu et al., 2019*). Numerous variables, such as air deposition from vehicle emissions or heavy traffic along the Hawassa-Moyale road, could cause heightened Cu concentrations (*Szwalec et al., 2020*). Furthermore, coffee species, geographic origin, coffee variation, usage of fertilizers with various chemical compositions, and other differentiating elements all have a substantial impact on the actual metal content of coffee beans (*Kamal et al., 2008*). The copper concentration levels reported in this study were below the permissible limits for agricultural land use of $300 \, \text{mg kg}^{-1}$ (*FAO & WHO, 2019*), implying that there was no Cu contamination in the soil.

In this study, the highest mean Cd concentration was ($3.36 \pm 0.1 \, \text{mg kg}^{-1}$), which is much lower ($124.3 \, \text{mg kg}^{-1}$) than that reported by a previous study (*Tezottoa et al., 2012*). However, it was higher than ($0.1 \pm 0.03 \, \text{mg kg}^{-1}$) reported in the agricultural land of Keniya (*Ndungu et al., 2019*) and Cd ($0.001 \pm 0.0009 \, \text{mg kg}^{-1}$) reported in the Western Ghats Region, India (*Raghavendra & Venkatesha, 2020*). On the other hand, the values reported for Cd ($3.49 \pm 0.26 \, \text{mg kg}^{-1}$) in a previous study (*Dubale, 2021*) are comparable with the results obtained in the present study. These differences might be due to the difference in anthropogenic sources, such as the application of fertilizers (*Raymond & Felix, 2011*). Cadmium levels reported in the soil are above the level of $3 \, \text{mg kg}^{-1}$, the permissible limit for agricultural soil (*FAO & WHO, 2019*). Therefore, there was Cd contamination in the soil.

Calcium had the highest concentration ($1,355 \pm 18.02$ mg kg$^{-1}$) of macro-elements in all soil samples, followed by K ($681.43 \pm 1.52$ mg kg$^{-1}$) and Na ($111.63 \pm 0.35$ mg kg$^{-1}$). Similarly, Cu ($49.96 \pm 0.99$ mg kg$^{-1}$) was detected in greater abundance among the microelements, followed by Mn ($0.62 \pm 0.238$ mg kg$^{-1}$) and Zn ($0.163 \pm 0.007$ mg kg$^{-1}$). Co>Cd>Ni was assigned to the remaining trace metals. The three main forms of Ca$^{2+}$ found in soils are as minerals, in solution, and attached to organic matter and clay minerals at exchangeable sites. The soil solution only contains a small portion of the total Ca$^{2+}$. Only Ca$^{2+}$ can be absorbed by plant roots from soil solutions (*Ramírez-Builes et al., 2020*). The kind of soil, the mineral composition of the colloids, the pH, the amount of organic carbon, the presence of humid acids, and the cation exchangeable capacities are just a few variables that may affect the amount of Ca$^{2+}$ that is present in the soil solution (*Ramírez-Builes et al., 2020*).

In soil samples from six kebeles, metals such as Pb and Cr were not detected. Kege Kebele soil sample analysis showed that Ca ($1,355 \pm 18.02$ mg kg$^{-1}$) was the highest of all the detected metals. K and Na were found in the highest amounts next to Ca, with values of $673 \pm 2.65$ mg kg$^{-1}$ and $111.63 \pm 0.35$ mg kg$^{-1}$, respectively. Of the analysed microelements, Cu ($42.66 \pm 2.52$ mg kg$^{-1}$) was found to be in the highest amount, followed by Co ($5.33 \pm 0.305$ mg kg$^{-1}$) and Mn ($0.62 \pm 0.238$ mg kg$^{-1}$). Likewise, the concentrations of Cd, Ni and Zn were found to be $3.1 \pm 0.1$ mg kg$^{-1}$, $0.191 \pm 0.009$ mg kg$^{-1}$ and $0.161 \pm 0.007$ mg kg$^{-1}$, respectively. The average metal concentrations in the kege soil were determined to be in the following order: Ca>K>Na>Cu>Co>Cd>Mn>Ni>Zn.

According to *Rakesh Sharma & Raju (2013)*, a high correlation coefficient (near $+1$ or $-1$) indicates a good relationship between two variables, whereas a concentration around zero indicates no relationship at a significance level of 0.05%. If $r > 0.7$, it is strongly correlated, whereas r values between 0.5 and 0.7 indicate a moderate correlation between two different parameters. This study revealed that there is a positive correlation between Cd and Zn, Ca, Na and K, indicating a linear relationship between the metals. The rise in levels of Cd increases the tendency of Zn, Ca, Na and K to increase. This is in line with previously reported data reported by *Dubale (2021)*, who found that Mg, Ca, Cr, Mn, Zn and Co in farmland soils in Gedeo, Ethiopia, may have a similar origin determined by a correlation study of soil heavy metals.

Potassium (K) was found to have the greatest concentration ($99.93 \pm 0.037$ mg kg$^{-1}$) among the macro-elements detected in coffee beans from all sampling farm land sites. As suggested by *Weis & Weis (2004)*, the highest concentrations of K in coffee beans are probably to be attributable to the truth that nutrients like K,N, P, S, and Mg are particularly cell in plant tissues and can be trans placed from ancient to younger plant tissues. According to *Çalişkan & Çalişkan (2017)*, another reason for increased K concentrations is because this element is among the most important nutrients for plants, and with the exception of N, potassium is the element that plants absorb the most compared to other elements.

The concentration of K in coffee beans from farmland reported in the present study ($99.93 \pm 0.037$ mg kg$^{-1}$) was much lower ($18,634.66 \pm 538.67$ mg kg$^{-1}$) than that reported in Poland (*Janda et al., 2020*) and K ($15,042.8 \pm 53.29$ mg kg$^{-1}$) from the Gedeo Zone, Ethiopia (*Dubale, 2021*). On the other hand, values reported for K ($21.31$–$427.84$ mg kg$^{-1}$)

by *Omer, Labib & Zafar (2019)* are comparable with the results obtained in the present study. The studied coffee beans were above the permissible limit set by FAO/WHO, which is 32,500 mg kg$^{-1}$ in plants. K is essential because it lowers the risk of stroke, supports healthy nerve and brain function, has diuretic qualities and helps control the water and acid–base balance in the blood and tissues. It has apparently been demonstrated that patients with high blood pressure can lower their blood pressure by eating a high-potassium diet (*He & Macgregor, 2008*).

The findings of the present study revealed that Na and Ca were also detected in significant amounts in coffee bean samples from farmers' farms, with concentrations of 22.04 ± 0.042 mg kg$^{-1}$ and 17.23 ± 0.36 mg kg$^{-1}$, respectively. This result is consistent with the Na concentration (6.84–564.74) and Ca concentration (6.76–32.09 mg kg$^{-1}$) reported in Saudi Arabia (*Omer, Labib & Zafar, 2019*; *Adler, Nkedzarek & Tórz, 2019*) Na concentration (18.6 ± 11.31 mg kg$^{-1}$) reported by in green coffee beans. However, the Ca concentration (17.23 ± 0.36 mg kg$^{-1}$) in the present study was much lower (1,252.93 ± 30.17 mg kg$^{-1}$) than that reported in the Gedeo Zone (*Dubale, 2021*). The Na concentration (22.04 ± 0.042 mg kg$^{-1}$) in the present study was much higher (1.8–9.3 mg kg$^{-1}$) than that reported in green coffee bean (*Martín, Pablos & González, 1998*). Such great differences in Na (6.1 ± 2.3 mg kg$^{-1}$) concentration were also reported in a previous study (*Grembecka, Malinowska & Szefer, 2007*). This variation might be due to the type of soil where the coffee was cultivated (*Santos & Oliveira, 2001*).

The highest concentration observed for manganese in coffee beans in the present study was 0.927 ± 0.004 mg kg$^{-1}$. When we compare the present experimental value with different countries previously reported. The findings of the present study were much lower (8.63 ± 10.14 mg kg$^{-1}$) than those reported in a previous study (*Omer, Labib & Zafar, 2019*) and reported Mn (23.60 ± 1.30 mg kg$^{-1}$) concentrations in the Gedeo Zone, Ethiopia (*Dubale, 2021*). However, it fairly agrees with the range (0.48–28.69 mg kg$^{-1}$) reported by (*Omer, Labib & Zafar, 2019*). The soil plant system is highly specific for different elements, plant species and environmental conditions (*Gure, Chandravanshia & Godeto , 2017*). As a result, coffee species, geographic origin, coffee kind, the application of fertilizers with varying chemical compositions, and other distinguishing characteristics all have a significant impact on the actual metal content of coffee beans (*Kamal et al., 2008*). Mn has a high solubility at low pH, and its concentration in acidic soil is likely to be high(*Wood, 1985*) The permissible limit set by *FAO/WHO (1993)* for edible plants is 2 mg kg$^{-1}$.

The highest concentration of Ni (0.074 ± 0.003 mg kg$^{-1}$) was reported in the present study, which is in good agreement with Ni (0.05–0.39 mg kg$^{-1}$) reported in coffee beans (*Grembecka, Malinowska & Szefer, 2007*) and Ni (0.07 ± 0.11 mg kg$^{-1}$) reported in coffee beans (*Omer, Labib & Zafar, 2019*). However, the results presented in the present study are generally lower (0.415 ± 0.04 mg kg$^{-1}$) than those reported in a previous study (*Adler, Nkedzarek & Tórz, 2019*) and the Ni (2.43 ± 0.14 mg kg$^{-1}$) concentration in coffee beans (*Dubale, 2021*). The steps used in the production of natural or soluble coffees, the variety and type of coffee, the methods used in the preparation and storage of coffee, and the origin (particularly the type of soil where coffee plants are grown) all have a significant impact on

the concentration of Ni (*Santos & Oliveira, 2001*; *Grembecka, Malinowska & Szefer, 2007*). The maximum permissible limit for Ni set by *FAO/WHO (1993)* for edible plants is 1.63 mg kg$^{-1}$.

The zinc concentration in the samples analysed ranged between 0.054−0.076 mg kg$^{-1}$, which was much lower than the zinc concentration (Zn) reported in Yemeni green coffee beans (3.74 to 46.89 mg kg$^{-1}$) (*Nogaim et al., 2014*), Zn reported in Brazil (*Silva et al., 2017*) (5.53 to 55.83 mg kg$^{-1}$), Zn reported in green coffee beans (3.09−4.04 mg kg$^{-1}$) (*Adler, Nkedzarek & Tórz, 2019*), and Zn reported in Brazil (8.74–12.75 mg kg$^{-1}$) (*Dubale, 2021*). However, this result is consistent with the Zn concentration (0.0−4.59 mg kg$^{-1}$) reported in Saudi Arabia (*Omer, Labib & Zafar, 2019*). Zinc concentrations vary greatly depending on a number of factors, including the coffee's origin, variety, and type coffee confection and storage *Ashu & Chandravanshi (2011)*.

In the present study, Co and Cr were not found in coffee samples from farmlands. This finding is consistent with the previous study reported by *Getachew & Worku (2014)* for raw and roasted coffee beans and *Santos & Oliveira (2001)* in Brazilian soluble coffee. However, a previous study reported Co (2.6−8.4 mg kg$^{-1}$) and Cr (0.21−0.28 mg kg$^{-1}$) concentrations in Ethiopian green coffee beans (Abera (*Gure, Chandravanshia & Godeto, 2017*). Similarly, another study reported Co (2.47–2.86 mg kg$^{-1}$), and Cr (1.04−1.92 mg kg$^{-1}$) concentrations in Ethiopian green coffee beans (*Dubale, 2021*). Co and Cr are mostly polluted by mining, manufacturing, fertilizers, pesticides, and rock weathering (*Zhou et al., 2020*). The negligent application of fertilizers, wastewater discharge, burning of coal and motor fuels, and increased mining of cobalt ore have all contributed to an increase in the concentration of naturally occurring cobalt (*Saaltink et al., 2014*). The maximum permissible limit for chromium set by *FAO/WHO (1993)* for edible plants is 0.02 mg kg$^{-1}$. However, there was no maximum permissible limit for cobalt set by *FAO/WHO (1993)* for edible plants.

The copper concentration in the present study ranged between 0.14−0.28 mg kg$^{-1}$, which is much lower (12–13 mg kg$^{-1}$) than the previous study reported by *Getachew & Worku (2014)*. *Gure, Chandravanshia & Godeto (2017)* reported Cu (11–23 mg kg$^{-1}$) for raw coffee beans; *Adler, Nkedzarek & Tórz (2019)* reported Cu (9.4–18.5 mg kg$^{-1}$) for green coffee; and *Dubale (2021)* reported Cu (23.39–28.5 mg kg$^{-1}$) coffee from farmland. However, the findings of the present study closely resemble the 0.13–10 mg kg$^{-1}$ range reported by (*Omer, Labib & Zafar, 2019*). Cu was found at low concentrations in samples of coffee from farmland. The reason for this low concentration is the soil where coffee plants grow and the environmental conditions that affect the concentration (*Ayele, Urga & Chandravanshi, 2015*). The concentration is affected by a variety of factors, including the type of coffee used, the fertilizer used, and the fertilized ground where crops were grown (*Suseela, Bhalke & Kumar, 2001*). *FAO/WHO (1993)* defined an acceptable level of 3.00 mg kg$^{-1}$ in edible plants. After comparing the metal limit in the investigated coffee beans to the *FAO/WHO (1993)* recommendations, it was discovered that all coffee beans collected Cu below this level. However, the *WHO (2005)* limits for Cu have not yet been determined for medicinal plants. Acceptable Cu limits in medicinal plants were determined by China and Singapore at 20 mg kg$^{-1}$ and 150 mg kg$^{-1}$, respectively *WHO (2005)*.

In coffee beans of the selected study site, Pb and Cd were not found. This finding is consistent with the previous study reported by *Getachew & Worku (2014)* for raw and roasted coffee beans, *Gure, Chandravanshia & Godeto (2017)* for raw coffee beans and *Dubale (2021)* for coffee bean samples from farmland. However, *Adler, Nkedzarek & Tórz (2019)* observed higher values for Pb ($0.076 \pm 0.0956$ mg kg$^{-1}$) in Bosnia green coffee and reported by *Nogaim et al. (2014)* Pb ($0.599$–$7.989$ mg kg$^{-1}$) in Yemeni green coffee beans. This indicates that coffee beans' chemical composition differs depending on the region in which they are grown. These hazardous metals were not detected in the present study, which might be due to the lack of environmental degradation due to industrial operations or chemical contamination from sources such as vehicles and pesticides in Dale Woreda. Furthermore, the absence of commercial fertilizers and pesticides for coffee plantations in Ethiopia may be evidenced by the low amounts of hazardous metals (*Gure, Chandravanshia & Godeto, 2017*). Pb and Cd have no nutritional value for humans, regardless of their very low concentrations. As a result, consumers will not be exposed to any health risks as a result of these hazardous substances in dale Woreda coffee beans. The maximum permissible limits for lead and cadmium set by *FAO/WHO (1993)* for edible plants are 0.43 mg kg$^{-1}$ and 0.21 mg kg$^{-1}$, respectively.

The amount of trace metals taken up by plants was calculated to determine the bio-concentration factor (BCF) of trace metals from soil to plant (*Welde Amanuel & Kassegne Berhanu, 2022*). The principal pathway for potentially hazardous metals to enter the food chain is by the movement and deposition of heavy metals from soil to edible parts of plants (*Sharma, Nagpal & Kaur, 2018*). The bio-concentration factor is a significant measurement for the pollution assessment of soils with the highest level of trace metals (*Welde Amanuel & Kassegne Berhanu, 2022*). When BCF is larger than 1, it means that the plant has the potential to store the metal under consideration for examination (*Barman et al., 2000*). The amount and types of heavy metals present, plant species, soil physicochemical properties, and other factors all affect how quickly heavy metals are transferred to and accumulated by plants at different rates (*Sharma, Nagpal & Kaur, 2018*; *Fonge et al., 2021*).

Mn had the highest bio-concentration factor/transfer factor (1.495) among the elements examined. This demonstrates that there is a possibility that the trace metal is derived from natural sources, is available for uptake, and has a low rate of metal retention in soil (*European Union, 2002*). The higher transfer factor of Mn in coffee beans can create a chance for higher human consumption. Mn influences plant function in addition to being harmful to human health. *Suresh, Foy & Weidner (1987)* observed that excess soil Mn disrupted stomatal function in plants (*Suresh, Foy & Weidner, 1987*). Mn also causes human lung injury, such as cough, bronchitis, and pneumonitis, along with lung damage (*Boojar & Goodarzi, 2002*).

In the analysed samples, Cu had the lowest transfer factor (0.0048), which is at the Magara locations (SS6). It was apparent that the TF of some metals (Cu) decreased when the plants were grown in soil with higher contamination (*Mirecki et al. , 2015*). It might become more tightly bound to the soil and alter the composition of the soil. This shows that the plant tissue absorbed most of the soil's trace metal concentration. The general pattern of trace metal transfer in Kege Kebele (SS1) was Mn>Zn>Ni >Na>K>Ca >Cu.

This clearly shows that the Mn bioaccumulation factors in the coffee samples were higher than those for the other metals.

The beans from Megara coffee washing industries, such as coffee beans from farmer's farms, had the maximum amount of K (77.93 $\pm$ 0.115 mg kg$^{-1}$), followed by Na (10.47 $\pm$ 0.058 mg kg$^{-1}$) and Ca (3.55 $\pm$ 0.114 mg kg$^{-1}$). In the washing industry's beans, Mn (0.92 $\pm$ 0.001 mg kg$^{-1}$) was the highest accumulated trace metal, followed by Cu (0.277 $\pm$ 0.011 mg kg$^{-1}$) and Zn (0.094 $\pm$ 0.004 mg kg$^{-1}$). Other critical trace metal concentrations found in coffee beans were Ni (0.074 $\pm$ 0.003 mg kg$^{-1}$). The results show that the concentrations of Co, Cr, Pb and Cd were not detected.

## CONCLUSIONS

Mean metal concentrations in the soil were determined to be in the following order: Ca>K>Na>Cu>Co>Cd>Mn>Ni>Zn. Except for Cd, all metals analyzed were below the permissible limit set by the FAO/WHO. Cadmium levels reported in the soil are above the level of 3 mg kg$^{-1}$, the permissible limit for agricultural soil (FAO/WHO, 2001). Therefore, there was Cd contamination in the soil.

Metal levels in coffee bean samples from farmers' farms are in the following order: K>Na>Ca >Mn>Cu>Ni>Zn. Metal levels were found to be K>Na>Ca >Mn>Cu>Zn>Ni in coffee beans from the washing plants. In both coffees, the levels of toxic metals (Pb and Cd) were not determined, and trace heavy metal levels were below the FAO/WHO maximum permissible limits. As a result, there is no health risk linked with the use of Dale Woreda coffee beans due to harmful and trace heavy metals. Mn had the highest bio-concentration factor/transfer factor among the elements examined. However, Cu had the lowest transfer factor. The general pattern of trace metal transfer in Kege Kebele (SS1) was Mn >Zn >Ni >Na>K >Ca >Cu. According to the findings of this study, there are permitted levels of macro- and trace elements in coffee beans from farmlands and washing plants. As a result, metal pollutants have no effect on the coffee grown in Dale Woreda. As a result, concerned bodies should make more marketing and raise awareness to increase and spread the recognition and consumption of Dale Woreda coffee beans in the national and international coffee market.

## ACKNOWLEDGEMENTS

We would like to thank Hawassa University and the Sidama Region Water Bureau for permitting us to conduct the lab from their laboratories. A very special thanks goes out to Mr. Alemayehu Timotiwos, who is the head of Sidma Regional State Peace and Security, for moral support for the study.

### Funding

The authors received no funding for this work.

## Competing Interests

The authors declare there are no competing interests.

## Author Contributions

- Yohannes Seifu Berego conceived and designed the experiments, performed the experiments, analyzed the data, prepared figures and/or tables, authored or reviewed drafts of the article, and approved the final draft.
- Solomon Sorsa Sota conceived and designed the experiments, analyzed the data, authored or reviewed drafts of the article, and approved the final draft.
- Mihret Ulsido conceived and designed the experiments, analyzed the data, authored or reviewed drafts of the article, and approved the final draft.
- Embialle Mengistie Beyene performed the experiments, authored or reviewed drafts of the article, and approved the final draft.

## Data Availability

The raw measurements are available in the Supplementary File.

## Supplemental Information

Supplemental information for this article can be found online at http://dx.doi.org/10.7717/peerj.14789#supplemental-information.

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
