# Peer review of "The contents of essential and toxic metals in coffee beans and soil in Dale Woreda, Sidama Regional State, Southern Ethiopia"

_PeerJ, doi:10.7717/peerj.14789_

## Round 0.1 · original submission · Major Revisions

We have two critical reviews of this work. The comments are quite extensive and need a full response. Please describe in more detail the methodology procedures. I await resubmission of the revised manuscript.

Reviewer 1 ·

Basic reporting

The authors describe the analysis of trace elements of cofee beans and soils related to cofee production. The review of literature is well done and the authors show important hypothesis, besides that there are a great interest in this study and a lack in local literature.

Experimental design

Unfortunately, the methodology is not sufficiently clear and required major revision, especially, in digestion procedure.

Validity of the findings

The authors do not describe any validation procedure for analysis. It is not possible discuss results based in a not validated method.
The statistic of data is well done and correct for this type of analysis. There are robust data, but not analytically validated.

Additional comments

I strongly suggest more clear methodology procedures.

Line 168: The concentrated acid concentration is required, for example, is HNO3 68% w/w?

Line 172: Is it %w/v?

The microwave system paraments are required, the temperature ramp, pression, model.. etc...

Line 177: This procedure need be clarified. The concentrations, volumes, temperature and time must be described.

Line 222 and in all text: do not use ppm, use mg kg-1.

Line 239 and in all text: Adjust the number of significant algarisms.

Table 1: Put the elements in alphabetical order.

Adjust the significant algarisms, for example, for Cd: 3.36 ± 0.10 instead 3.36 ± 0.1; 2.67 ± 0.06 instead 2.67 ± 0.057.

Validation step need be performed. For this, read the guidance of validation in: “B. Magnusson and U. Örnemark (eds.) Eurachem Guide: The Fitness for Purpose of Analytical Methods – A Laboratory Guide to Method Validation and Related Topics, (2nd ed. 2014). ISBN 978-91-87461-59-0. Available from www.eurachem.org.”

·

Basic reporting

Overall comments on whole paper
The paper presents the contents of essential and toxic metals in green coffee beans collected from farms and washing plants and soils in coffee farms in Dale Woreda (district) of the Sidama Regional State, southern Ethiopia. The coffee samples were collected from farms at 6 kebeles and 6 pulping stations (coffee washing plants) at 1 kebele. The soil samples were taken from same farms that the coffee samples were collected and composited per kebele. The study is interesting from human health points of view.
However, in this paper, the following major issues of the scientific paper are missed:
1. No clear rationale of the study, especially for that of the toxic metals as possible situations that can influence the contents of toxic metals in soils and crop products including coffee beans are not known so far in the study area of this paper.
2. No clear nobility of the study as Bubale, W. (2021) carried out a similar study already in the same region.
3. No sufficient literature review on studied variables, clear experimental design and procedures of sample collection, processing and lab and data analyses
4. No clear and sufficient description (interpretation) and discussion of the results
5. The language in the entire paper is not to the standard of the scientific paper.

. Introduction:
- It doesn’t show rationale and/nobility of the study as well as clear definition of essential and toxic (heavy) metals. The objective of the study is not also complete/clear enough.
- Except some points in Lines 84-89, no sufficient literature review on the contents of essential and toxic (heavy) metals in green coffee beans and transfer of these metals from soil to the coffee beans or other crop products, such as cacao, grains and the like.
- Lines 62-65: is information for all developing countries/Are all developing countries coffee producers or exporters? Does coffee currently contribute 60% of the export earnings of Ethiopia?
- Lines 65-67: is Ethiopia the fifth coffee exporter in the world?
- Line 69: how do you suddenly conclude the vitality of assessing quality of coffee of Ethiopia?
- Lines 70-74: why did you bring these issues into your introduction? Are they a problem in your study area (in Sidama area) as well as in Ethiopia?
- Lines 55, 57, 59, 62: please identify those words that need to be started with capital or small letter;
- Lines 81: use either the symbol or full spell of each metal and include year of publication the Reference Suresh et al.
- Line 84: the phrase “our favorite brews” stands for which brews?
- Line 88: the phrase “These metals” stands for which metals?
- Lines 94-98 and Lines 101-103: what the authors want to say in these statements is not totally clear?
- Lines 103-105 (the objectives): require revision

Experimental design

- In the study area description, some relevant information on agricultural practices (e.g., application of agrochemicals, such as fertilizers and pesticides) and climates of the area (e.g., rainfall) that can influence soil chemical properties and elemental contents of coffee beans is missing
- In sample collection section, important information that can show the quality of the design of the study, such as number of coffee farms (coffee samples) per kebele, the number of coffee samples per coffee washing plants (pulping stations), why washing plants were selected only from one kebele and why one composite soil sample per kebele was preferred rather some representative samples per kebele as well as methods of sampling of the farms at each kebele and coffee at each washing plant is missing
- Lines 110: the geographical location shows a single place in the study area, but this study had six places (kebeles) in Dale district.
- Line 128: what does it “clean and washed bags?? What type of bags is this?
- Lines 142-146: How was coffee samples taken from washing plants and how were they processed? How the husk was removed?
- Line 151: detail analytical procedures of each metal in the soil is required
- Line 152 and 178: why question mark is used?
- Line 169: what are the optimum conditions for sample digestion? Need to specify?
- Line 208-217: data analysis is totally unclear. What statistical analysis was done by using each MS excel and SPSS? This needs to be specified. How ANOVA was done for metal contents in soil samples while there was only one composite sample kebele (when there was no replication). There is also a similar question on ANOVA of metal contents in coffee beans while there is no information on the number of coffee samples (replicates of coffee samples) per kebele and per washing plant as well as comparing the differences between coffee samples from farms and washing plants. What type of correlation was done between metal contents in soils and coffee beans?
- Lines 217-2018: information in this statement is not relevant here.

Validity of the findings

1. Results.
- In general, the description of the results is unnecessarily in detail (especially that under metal conc. in soil samples, Lines 221-286) and it is not based on the statistical results of the study.
- Lines 280-282: How did you say this? Are all these heavy metals?
- Lines 285-286: How did you say this from linear correlation analysis?
- Lines 306-308, 309-310 and 319-320: is the information in these statements correct?
- Lines 323-324: this is part of the data analysis
- Lines 280-282: need to present first those significantly affected results and following by non-significant ones.
Tables (data presentation):
- In Tables 1 and 6, need to reshuffle (exchange) the heading of the columns (kebeles) with that of the rows (metals) and follow the usual style of letter designation for statistically differed results; i.e., the letter ‘a’ is assigned for significantly higher results, followed by ‘b’ for the next higher results and so on
- But, there is a question on letter designation; how the letter designation can be given for the results presented in Table 1 while there was only one soil sample per kebele, and but not for the results presented in other Tables like Tables 3, 4, 5, and 6?
- Are all metals in Table 2 heavy metals? See the caption of this table? What is the relevance of the correlation analysis between metals in soils for this study (Table 2)? But, in the data analysis section (Lines 213-215), it is mentioned that the correlation analysis was carried between metals in soils and coffee beans. Which one is correct?
- Where are the results of essential metals (which are indicated in the title and objective of the paper) presented? Is an essential metal synonymous with a non-toxic metal?
- ND in Table 1; ND and MPL in Tables 3, 4 and 7; ND, NR, CBWI and CBFF in Table 8 (presented as Table 1 in draft MS); and CI in Table 9 are not described.
- No units of measurement for the data in all Tables and description for those figures after ‘±’ sign in captions of Tables 3, 4 and 6.
- Why was there no SD for the data in Table 5? What is the relevance of presenting equal variances in Table 9 and reporting the statistical information presented in Table 9? Is there no other better way to present the information presented in this Table?
2. Discussion
- It is too long (ca. 7 pages) and unclear. Moreover, the link between the findings of the study with human health and agricultural practice in the study area is not discussed.
- Lines 329-330: What can be the reasons for this difference between your results and those reported by Dubale?
- Lines 329-330: What about geological differences?
- Lines 329-330: What is the relationship between Ni and manure?
- Lines 349-351: How these factors indicated in this statement can be mentioned as reason for the conc. of heavy metals in study area of the present study?
- Line 352: the reference “Basta et al., 2005” is not about livestock manure.
- Lines 329-330: how could be the reason for your results? Where have all these been happened? Is it in your study area?
- Lines 368-369: Is Cu available in dusts and vehicle emissions?
- Lines 370-371: Are sure the existence of differences between your study sites (kebeles) in parent rocks, geography and climate?
- Lines 374-383: Why didn’t you discuss on Ca, Mn, Na, K, Cr, and Pb results like you did for others, such as Cd here?
- Lines 393-3395: Why did you bring a correlation in between the comparison the conc. of each metallic element in your stud and previous studies and their reasons?
- Lines 421-422: Do the findings of these studies go with those of your study?
- Lines 431-433: How do we know the situation observed in this sentence?
- Line 458: is it in soil or coffee samples? How did say this while Co was identified/quantified in soil samples of Kege and Wenenata kebeles?
- Why didn’t you discuss your results for coffee bean samples from washing plants?
- Lines 461 and 491: Are the Citations “Abera Gure” and such a citation style like “according to (Bubale, 2021)” and “by (Bubale, 2021)” all over the paper correct?
- Line 524: why cu had the lower transfer factor than other metals
3. Conclusion
- It is not clear enough and in proper order.
- Why no conclusion for results of bioconcentration factor?
- Lines 532-539: Are these relevant in the conclusion?
- Line 546: was statistical analysis done for coffee samples from washing plants

Additional comments

1. Title and authors affiliation:
- I would recommend to (1) modify the title to a statement that show the content (e.g., The contents of essential and toxic metals….) and the location of the study (e.g., southern Ethiopia) than the current one that shows the process of the study (i.e., determination) and regional state, and (2) properly write the affiliation of the first author (i.e., Biology) and remove the phrase “Sidama regional state from affiliations.
- Why does the first author have two different names and affiliations in Lines 6 and 17?
2. Abstract:
- No rationale of the study is indicated
- The methodology part is not clear enough
- Line 25: from where soil samples were collected? What is FAAS?
- Line 27: which samples (those of coffee or soil) were prepared in Microwave system?
- Lines 31-37: How (in what base) the metallic elements (n = 11) analyzed in this study were divided into macro and micro elements is not clear. Is Na a micro element in soils?
- Lines 37-40: How ANOVA was done from single composite soil sample per kebele to compare the 6 kebles in metal contents of their soils?
- Line 52: What are these FAO/WHO maximum permissible limits? Why the authors limited their conclusion only on results of absence/presence of toxic metals in coffee beans and the contents of trace heavy metals in relation to FAO/WHO maximum permissible limits?
3. General question:
- Why didn’t you include the conc. of the studied metals (especially those of heavy metals in waters released from washing plants and environmental issues in your study along with human health?

---

## Round 0.2 · Minor Revisions

There are no more remarks from the reviewers.
However some technical and logical points should be clarified in the manuscript before the publication. I add my comments as the editor:

Please rephrase in the Abstract:
Line 28: “Out of the 32 Kebele Administrations in Dale Woreda, only 20% were selected, which was six Kebele (smallest administrative unit).” - Kebele is new word, it is used in low case and upper case, maybe not clear to the readers. And logically it is not clear why 20% of the units were used.
Rephrase like “We selected 6 (20%) of administrative units (Kebele) in Dale Woreda for the soil testing”.
Conclusion section of the Abstract should be restructured:
Line 52-55: “In the soil samples, cadmium concentrations are higher than those permitted for agricultural soil recommended by WHO and FAO. There are permitted levels of macro and trace elements in coffee beans from farmlands and washing plants. As a result, there is no health danger...”

If “cadmium concentrations are higher” why “no health danger”?

Rearrange this part like this:
“We observed permitted levels of macro and trace elements in coffee beans from farmlands and washing plants. Only in the soil samples, cadmium concentrations are higher than those permitted for agricultural soil recommended by WHO and FAO. Overall, there is no health danger...”

In the Introduction part (lines 58-63) need a phrase about importance of biosafety estimates for the coffee production. Some text about the coffee cultivars is not relevant to the study.

Line 356: “manganese > zinc > nickel > sodium> potassium > calcium> copper “ - please make writing similar to the main text using chemical element notation.
Make the designations uniform in the main manuscript text.

Please check grammar, use proper punctuation in the text and citations
(for example, line 400 - “(Dubale, 2021) .However,...” - space before the dot sign
“Nigeria.(Oladeji and Saeed, 2015).” - extra dot before the citation
“Woreda, .Furthermore” - extra dot or comma?)

Please format the references in the reference list (remove uppercase for the authors’ names, make it in the same font, refresh access date to the web-links (“(accessed 13 November 2021). - too old”)

Reviewer 1 ·

Basic reporting

The authors satisfactorily answered the reviewers' questions and made the necessary modifications. I only suggest that you take care to correctly express the units mg kg-1(-1 superscript), correct this throughout the text.
Furthermore, I believe the manuscript can be published as is.

Experimental design

.

Validity of the findings

.

Additional comments

.

---

## Round 0.3 · Minor Revisions

Thanks for the manuscript update. All the critical remarks were considered.

However, there are a lot of typos (you use commas, dots, no spaces, or extra spaces at the end of the sentences, and so on). You need to check the grammar, presentation of formulas.

An important note is about additional references on the documents and standards allowing the presence of metal and chemical components in food products. You need to add such references to the text. See comments from the Section editor:

“Since the analysis focused on toxic metals it would be wise to include a citation highlighting what the accepted limits for potential toxicants might be, as international organizations were also cited. The manuscript may be sufficiently adequate; however, it may require some additional proofing and some attention to appropriate citations to make this less political. It is more of a sample test report than an experiment; however, may have some merit.”

I have attached a PDF file with the editor's comments and remarks.

Please check it in final revised version.

---

## Round 0.4 · accepted · Accept

Thank you for the manuscript update. There are no more critical comments. I recommend accepting this manuscript for publication now.

Happy New Year!